# Evaluation of an Image-Derived Input Function for Kinetic Modeling of Nicotinic Acetylcholine Receptor-Binding PET Ligands in Mice

**DOI:** 10.3390/ijms242115510

**Published:** 2023-10-24

**Authors:** Matthew Zammit, Chien-Min Kao, Hannah J. Zhang, Hsiu-Ming Tsai, Nathanial Holderman, Samuel Mitchell, Eve Tanios, Mohammed Bhuiyan, Richard Freifelder, Anna Kucharski, William N. Green, Jogeshwar Mukherjee, Chin-Tu Chen

**Affiliations:** 1Department of Radiology, University of Chicago, Chicago, IL 60637, USA; 2Fermi National Accelerator Laboratory, Batavia, IL 60510, USA; 3Department of Neurobiology, University of Chicago, Chicago, IL 60637, USA; 4Marine Biological Laboratory, Woods Hole, MA 02543, USA; 5Department of Radiological Sciences, University of California, Irvine, CA 92697, USA

**Keywords:** kinetic modeling, nicotine, addiction, PET, Nifene, 2-FA85380

## Abstract

Positron emission tomography (PET) radioligands that bind with high-affinity to α4β2-type nicotinic receptors (α4β2Rs) allow for in vivo investigations of the mechanisms underlying nicotine addiction and smoking cessation. Here, we investigate the use of an image-derived arterial input function and the cerebellum for kinetic analysis of radioligand binding in mice. Two radioligands were explored: 2-[^18^F]FA85380 (2-FA), displaying similar pKa and binding affinity to the smoking cessation drug varenicline (Chantix), and [^18^F]Nifene, displaying similar pKa and binding affinity to nicotine. Time–activity curves of the left ventricle of the heart displayed similar distribution across wild type mice, mice lacking the β2-subunit for ligand binding, and acute nicotine-treated mice, whereas reference tissue binding displayed high variation between groups. Binding potential estimated from a two-tissue compartment model fit of the data with the image-derived input function were higher than estimates from reference tissue-based estimations. Rate constants of radioligand dissociation were very slow for 2-FA and very fast for Nifene. We conclude that using an image-derived input function for kinetic modeling of nicotinic PET ligands provides suitable results compared to reference tissue-based methods and that the chemical properties of 2-FA and Nifene are suitable to study receptor response to nicotine addiction and smoking cessation therapies.

## 1. Introduction

Tobacco use is the leading cause of preventable deaths in the United States and one of the prominent causes of nicotine addiction [1]. Nicotine permeates the blood–brain barrier and binds to high-affinity nicotinic acetylcholine receptors (nAChRs) containing α4 and β2 subunits (α4β2Rs) [2]. Chronic exposure to nicotine causes upregulation of α4β2Rs, in which increases in both the density of high-affinity binding sites and the functional response of α4β2Rs are observed [3,4,5,6]. In addition, the process of nicotine-induced α4β2R upregulation has been linked to nicotine addiction [7,8].

Nicotine, and other weak-base ligands of α4β2Rs, such as the smoking cessation drug varenicline (Chantix), rapidly reach equilibrium in intracellular organelles and concentrate in acidic organelles [9,10]. The high pKa and binding affinity of varenicline causes selective trapping of this ligand inside intracellular acidic vesicles containing high-affinity α4β2Rs [10]. Alternatively, nicotine concentrates inside these acidic vesicles but does not become trapped due to its lower pKa and lower binding affinity and is rapidly released from the vesicles [10]. Nicotine-induced upregulation increases the number of high-affinity binding sites within the acidic vesicles and increases the number of acidic vesicles, allowing for the vesicles to trap varenicline in higher concentrations. Recently, our in vitro studies suggest these acidic vesicles to be Golgi satellites (GSats), a novel intracellular compartment in neurons and neuronal dendrites that contain high α4β2R density, which increase in number following exposure to nicotine [10,11]. While nicotine and varenicline bind to the same α4β2Rs, the residence time of nicotine in the brain is 1–2 h, compared to the 4–5-day residence time of varenicline, which may be attributed to varenicline trapping inside GSats. It has been shown that dissipating the pH gradient across GSats with chloroquine diphosphate or ammonium chloride prevents trapping of varenicline in GSats, and under these conditions, exposure to varenicline results in similar α4β2R upregulation as observed with nicotine [10]. Cell pretreatment with chloroquine diphosphate or ammonium chloride did not affect the extent of α4β2R upregulation induced by nicotine exposure.

Of interest is to monitor the effects of nicotine addiction and smoking cessation in vivo. This can be achieved using positron emission tomography (PET), in which the binding of nicotinic ligands to α4β2Rs can be observed noninvasively through injection of nanomolar concentrations of radiolabeled nicotine analogs [12]. Initial PET studies imaged [^11^C]nicotine; however, this ligand suffered from rapid dissociation of the receptor–ligand complex, high levels of nonspecific binding, and its accumulation in the brain was highly dependent on cerebral blood flow [13,14,15]. The radioligand 2-[^18^F]FA85380 (2-FA) was developed as a less toxic analog of epibatidine that overcomes the shortcomings of [^11^C]nicotine and binds with high affinity to α4β2Rs [16], but requires a prolonged imaging session to achieve accurate quantification [17]. [^18^F]Nifene was then developed as a ligand with moderate affinity to α4β2Rs to improve upon the slow kinetics of 2-FA [18]. Since the slow kinetics of 2-FA closely resemble epibatidine and varenicline, and the fast kinetics of Nifene resemble nicotine, these ligands can be used to monitor the mechanisms of smoking cessation and nicotine addiction in vivo. Our previous in vitro findings with these ligands found that chronic exposure to Nifene resulted in similar α4β2R upregulation as observed with nicotine, whereas exposure to 2-FA did not cause upregulation, similar to varenicline [10]. The high pKa and high affinity of 2-FA likely result in the same GSat trapping observed with varenicline, and pH dissipation across GSats with chloroquine diphosphate or ammonium chloride prevented 2-FA trapping and caused significant α4β2R upregulation following exposure to 2-FA. This phenomenon was also observed in vivo using PET, in which mice pretreated with chloroquine diphosphate showed reduced binding of 2-FA, while Nifene binding was unaffected [10].

One challenge associated with PET quantification of 2-FA and Nifene is the lack of a suitable reference tissue due to the abundance of α4β2Rs in the brain. Typically, the cerebellum is chosen as a reference tissue due to its low uptake and rapid washout of α4β2R-binding radioligands. In nonhuman primates, the cerebellum was validated as a suitable reference tissue for α4β2R-binding radioligands [17]; however, in rodents, radioligand concentrations in the cerebellum can be displaced by nicotine or lobeline injection [19], indicative of some specific binding signal. Human imaging studies incorporated the corpus collosum as a reference tissue; however, this needs to be further validated using a nicotine challenge to measure the nicotine displaceable component [18]. It is speculated that the corpus collosum may be a suitable reference tissue free of specific binding in rodents, but partial volume effects may negatively affect quantification in this region due to its small volume and the low spatial resolution of PET. Due to the lack of a true tissue reference region for rodent imaging, studies incorporating the cerebellum as a reference tissue likely underestimate the true binding potential. Apart from performing arterial cannulation to measure the input function, use of an image-derived input function for kinetic radioligand analysis is speculated as a noninvasive alternative to improve image quantification [20]. Studies incorporating image-derived input functions have shown that the time-activity curve (TAC) from the left ventricle of the heart was an accurate representation of arterial blood, obviating the need for arterial cannulation [21,22,23,24,25].

Here, we provide an exploratory analysis of the use of the left ventricle as an image-derived input function without arterial sampling for quantification of 2-FA and Nifene PET images of mice, as well as use of the cerebellum as a tissue reference region. As preclinical PET images of rodents encompass the entire animal body within the scanner field of view, obtaining a TAC from the left ventricle is feasible for kinetic PET analyses. This study is the first to explore use of an image-derived input function for the quantification of nicotinic PET ligands in preclinical mouse models. Left ventricle TACs were compared between wild type, β2-knockout, and acute nicotine-treated mice to explore how different mouse models influence radioligand activity concentration in the blood pool. Using a two-tissue compartment model fit (2TCM) of the PET data, rate constants of radioligand association and dissociation were calculated to determine the radioligand binding potential. Binding potential values were then directly compared against estimates derived using the cerebellum reference tissue previously explored for these radioligands. Finally, 2TCM fit simulations were performed to assess the stability of the rate constant estimates using the left ventricle TAC input function.

## 2. Results

### 2.1. Left Ventricle Comparisons

2-FA and Nifene time-activity curves (TACs) of the left ventricle, thalamus, midbrain, and cerebellum for all mice are provided in Figure 1a. For 2-FA, peak SUVs in the left ventricle (presented as mean (SD)) for WT mice (0.55 (0.17)), KO mice (0.47 (0.22)), and AN mice (0.569 (0.12)) showed no significant differences (ANOVA F(df): 0.62(2); *p*-value: 0.54; η^2^: 0.05 [0.00, 0.23]). Mean SUVs from the final 30 min data frames of WT mice (0.077 (0.029)), KO mice (0.081 (0.049)), and AN mice (0.074 (0.014)) showed no significant differences (ANOVA F(df): 0.75(2); *p*-value: 0.93; η^2^: 0.006 [0.00, 0.0.071]). For Nifene, peak SUVs in the left ventricle (presented as mean (SD)) for WT mice (0.98 (0.21)), KO mice (1.4 (0.95)), and AN mice (1.2 (0.15)) showed no significant differences (ANOVA F(df): 1.38 (2); *p*-value: 0.27; η^2^: 0.11 [0.00, 0.32]). Mean SUVs from the final 30 min data frames of WT mice (0.41 (0.14)), KO mice (0.45 (0.14)), and AN mice (0.50 (0.12)) showed no significant differences (ANOVA F(df): 0.64(2); *p*-value: 0.54; η^2^: 0.05 [0.00, 0.22]).

### 2.2. Binding Potential Comparisons

Using the left ventricle TAC as an input to the 2TCM, *BP*_ND_ values were calculated for the thalamus, midbrain, and cerebellum of WT, KO, and AN mice by taking the ratio of *k*_3_ and *k*_4_ (Figure 1b). From ANOVA with post hoc Tukey’s HSD, 2-FA *BP*_ND_ values were significantly higher in the thalamus, midbrain, and cerebellum of WT mice compared to KO and AN mice (*p* < 0.05), while no significant difference was observed between KO and AN mice (*p* > 0.05). For Nifene, *BP*_ND_ values were significantly higher in the thalamus, midbrain, and cerebellum of WT mice compared to KO and AN mice (*p* < 0.05), while no significant difference was observed between KO and AN mice (*p* > 0.05).

### 2.3. 2TCM Rate Constants

Table 1 provides the rate constant estimates with ANOVA F-statistics and effect size estimates (η^2^ with 95% CIs) from the 2TCM for the thalamus and midbrain imaged with 2-FA and Nifene. For 2-FA, significant differences were observed between WT, KO, and AN mice for *K*_1_ values in the thalamus, but not the midbrain. No significant differences were observed with *k*_2_ values in both the thalamus and midbrain across all groups. Significant differences between groups for both regions were observed for *k*_3_ and *k*_4_ values. Post hoc Tukey’s HSD revealed that the WT mice had higher estimates of *k*_3_ and lower estimates of *k*_4_ compared to KO and AN mice. In addition, Tukey’s HSD revealed no significant difference in *k*_3_ and *k*_4_ values between KO and AN mice. For Nifene, significant differences were observed between mice for *K*_1_ values in the thalamus and midbrain. From Tukey’s HSD, the significance was driven by the comparison between WT and KO mice. KO and AN mice and WT and AN mice showed no significant difference between *K*_1_ estimates. No significant difference in *k*_2_ values were observed between mice in the thalamus, but midbrain *k*_2_ values were significantly different. For the midbrain, *k*_2_ values in WT mice were significantly higher than in the KO mice, and no differences were observed between the other pairings of mice. Significant difference in *k*_3_ values were observed between mice in the thalamus and midbrain. For both the thalamus and midbrain, no significant differences were observed for *k*_4_ values between mice.

### 2.4. Rate Constant Comparisons between In-House Python Solver and PMOD Solver

Table 2 displays the 2-FA rate constant estimates from the Python and PMOD 2TCM solvers, and the pairwise *t*-test comparisons between them. For WT mice, there were no significant differences between estimates of *K*_1_, *k*_2_, *k*_3_, or *k*_4_ in the thalamus or midbrain between methods. PMOD estimates for *K*_1_, *k*_3_, and *k*_4_ were significantly higher than the Python solver in KO mice, and PMOD estimates for *k*_3_ and *k*_4_ were higher in AN mice. Since KO and AN mice display no specific binding of 2-FA, estimates of these rate constants display high variance, which may influence the trend of higher mean values. Table 3 displays the Nifene rate constant estimates from the Python and PMOD 2TCM solvers, and the pairwise *t*-test comparisons between them. For WT mice, there were no significant differences between estimates of *k*_3_ or *k*_4_ in the thalamus or midbrain between methods. However, *K*_1_ and *k*_2_ estimates from the Python solver were significantly higher than the PMOD estimates. KO and AN mice revealed no significant differences between rate constant values in the thalamus and midbrain. Lack of Nifene-specific binding in KO and AN mice contributes to the higher variance in these measurements for the Python and PMOD solvers.

### 2.5. 2TCM Comparisons to Logan Graphical Analysis

The 2TCM estimate of *BP*_ND_ + 1 from the Python solver using a left ventricle input function was directly compared to the Logan graphical analysis estimate of DVR using a cerebellum reference tissue. Table 4 displays the pairwise comparisons between *BP*_ND_ + 1 and DVR. For 2-FA, *BP*_ND_ + 1 values were significantly higher than DVRs in the thalamus and midbrain of WT mice. KO and AN mice also displayed significantly higher *BP*_ND_ + 1 values compared to DVR; however, the values are low and indicative of no specific binding signal. As illustrated in Figure 1b, *BP*_ND_ in the cerebellum of WT mice was significantly higher than observed in KO and AN mice, indicative of some 2-FA specific binding in this region. This cerebellar specific binding signal results in an underestimation of the true binding potential in the thalamus and midbrain when estimated from Logan DVR. For Nifene, no significant difference between *BP*_ND_ + 1 and DVR were observed in the thalamus or midbrain of WT mice, likely because the levels of cerebellar specific binding are much lower compared to 2-FA as shown in Figure 1b and further illustrated by the SUV TACs in Figure 1a. Nifene *BP*_ND_ + 1 in KO and AN mice also displayed higher values compared to DVR; however, these values are low and indicative of no radioligand specific binding.

### 2.6. Simulations

Simulations were performed on the PET data to evaluate the stability of the output parameters of the 2TCM fit when solved using the in-house Python solver. Briefly, the 2TCM fit was used to determine the true values of the parameters from the regional TACS from a WT, KO, and AN mouse. For each time point of the TAC, artificial noise was applied to the radioactivity concentration value with a noise level of 0.2 (noise standard deviation relative to the amplitude) such that the noise level was proportional to the amplitude/time duration. The 2TCM fit was then applied to the noise-induced thalamus and cerebellum TACs using the noise-induced left ventricle TAC as the input function. This process was repeated over 50 iterations for each mouse and radioligand to solve for *BP*_ND_, *K*_1_, *k*_2_, *k*_3_, and *k*_4_ values for the WT, KO, and AN mice. Figure 2 displays the results of the simulated 2-FA data. For the WT and KO mice, the true values of *BP*_ND_, *K*_1_, *k*_2_, *k*_3_, and *k*_4_ all fell within the 25th and 75th percentile of the simulated estimates. For the AN mice, the true values of *BP*_ND_, *K*_1_, *k*_2_, *k*_3_, and *k*_4_ all fell within the 25th and 75th percentile of the simulated estimates, except for the *BP*_ND_ of the thalamus. Figure 3 displays the results of the simulated Nifene data. For *BP*_ND_ values, the true values fell within the 25th and 75th percentile of the simulated estimates for the WT, KO, and AN mice. For *K*_1_ and *k*_2_ values, the 2TCM fit slightly underestimated the true values for all groups of mice. For *k*_3_ and *k*_4_ values, the true values fell within the 25th and 75th percentile of the simulated estimates for the KO and AN mice, while the WT mice simulations slightly overestimated the true values.

## 3. Discussion

This study is the first to compare an image-derived input function with tissue reference region strategies for kinetic analysis of 2-FA and Nifene PET images in mice. An ROI of the left ventricle was chosen as the image-derived input function due to its accurate representation of the plasma fraction as shown in other studies [20,21,22,23,24,25], and in its current application, the left ventricle TAC showed rapid uptake and rapid clearance of 2-FA and Nifene in the blood pool. Across WT, KO, and AN mice, no differences in shape or radioactivity concentration were observed between the SUV TACs of the left ventricle, suggesting the left ventricle blood curve is stable across different mouse models studied in nicotine addiction and can be a suitable reference for these mouse models when imaging with 2-FA and Nifene. Previous studies using α4β2R PET ligands in rodents utilized the cerebellum as a tissue reference region and identified nicotine-displaceable signals in this region [19]. The current study confirms the presence of α4β2Rs in the cerebellum, notably due to the higher binding potential values of 2-FA and Nifene in the WT group compared to the KO and AN groups. Thus, studies with rodents using the cerebellum as a tissue reference region will underestimate the true binding potential in target regions of interest, further emphasizing the need for improved quantification derived from arterial data. However, the low level of specific binding in the cerebellum would only slightly underestimate the true binding potential. Importantly, DVR estimates from Logan graphical analysis using the cerebellum as a reference tissue showed lower standard deviation across the groups of mice compared to *BP*_ND_ derived from the 2TCM with the image-derived input function. In clinical settings with smaller sample sizes, it may be advantageous to use brain tissue reference regions to minimize the variance across individual estimates. One limitation to our application of using an image-derived input function was delivery of the radioligands through IP injection. IP injections result in a slower distribution of radioligand from the plasma compartment to target regions when compared to the more standard IV injection. As a result, 2TCM fitting of the radioactivity time course of Nifene, which has very rapid kinetics in vivo, was challenging as the data were better represented by a 1TCM. The goodness of fit of the models, determined through the Akaike Information Criterion (AIC) confirmed that the 2TCM performed better for 2-FA (mean AIC across all mice = −988 (98)) compared to the 1TCM (AIC = −922 (143)). For Nifene, the 1TCM (AIC = −2006 (293)) outperformed the 2TCM (AIC = −1952 (303)). Compared to 2-FA, Nifene is more lipophilic and tends to concentrate in abdominal fat when injected IP, resulting in a slower plasma time course. Since the kinetics of 2-FA are very slow, use of an IP injection did not present a challenge in 2TCM fitting. Despite the challenges associated with IP injections, the 2TCM fit of the data using an image-derived input function resulted in suitable estimates of the binding potential for 2-FA, while a 1TCM fit was suitable for Nifene.

While this study is the first to assess kinetic modeling strategies of α4β2R ligands in mouse models of nicotine addiction, other studies have evaluated the kinetics in nonhuman primates [26,27]. For rhesus macaques, modeling of dynamic Nifene PET data was performed with arterial sampling and metabolite correction [26,27]. *BP*_ND_ values in cortical and subcortical regions of known Nifene binding were comparable between the rhesus macaques [26] and the mice used in our current study. Another study in rhesus macaques performed a 2TCM fit of the PET data with metabolite correction and found that average *K*_1_ values for Nifene were above 1.0 (1/min), confirming the high *K*_1_ values observed in our mouse models without arterial sampling and metabolite correction. *K*_1_ values that exceed 1.0 (1/min) indicate complete extraction of the radioligand from plasma to tissue, consistent with what is observed with Nifene [27]. It is speculated that a transport mechanism is at play, specifically Nifene interaction with the blood–brain barrier amine transporter [28], which may account for the fast uptake rates of Nifene compared to 2-FA [27]. Similar α4β2R ligands with rapid in vivo kinetics, such as Flubatine (formerly NCFHEB), have been shown to interact with the blood–brain barrier amine transporter [29]. Furthermore, work in the field is ongoing to investigate the role of the amine transporter on Nifene passage across the blood–brain barrier [27].

The current study incorporated use of an in-house Python solver and PMOD for the 2TCM fits of the PET data. The in-house Python solver was developed to derive more stable estimates of the rate constants for the KO and AN mouse groups. Due to the low binding levels of 2-FA and Nifene in these groups, rate constant estimates using PMOD had high uncertainties. Use of parameter optimization in the Python solver greatly reduced these uncertainties in the estimates, allowing for better comparisons to the rate constants derived for the WT mice. Importantly, estimates calculated from the Python solver and PMOD were of within the same order of magnitude when fitting data from the WT mice, which displayed high levels of radioligand binding. For the WT mice, estimates of *k*_3_ and *k*_4_ were large for Nifene, indicative of rapid binding and unbinding from α4β2Rs. Alternatively, 2-FA showed small estimates of *k*_3_ and very small estimates of *k*_4_, indicative of slow binding and very slow unbinding from α4β2Rs. This very slow unbinding of 2-FA may partially be influenced by ligand trapping inside GSats, which has been shown previously in vitro and in vivo [10,11,30]. Of interest is to use these radioligands to directly study the mechanisms of nicotine addiction and smoking cessation. In vitro cellular studies performed under similar in vivo conditions in mice found that the dissociation rate of nicotine to be 0.84 1/min [31], while the dissociation rate of epibatidine was 0.043 1/min [32]. The in vivo dissociation rate (*k*_4_) estimates of Nifene and 2-FA derived in the current study (Table 1) fall within the same order of magnitude as the in vitro estimates of nicotine and epibatidine, suggesting that these PET ligands may be useful to study addiction and smoking cessation mechanisms, especially those involving GSat trapping and release. Because 2-FA was only imaged for a 3-h duration, a true estimate of 2-FA release from GSats could not be obtained, as radioligand did not reach equilibrium in the brain by the end of the scan. Previous in vitro work measured the dissociation of epibatidine over a 17 h duration and found that there was a rapid component (unbinding from α4β2Rs) and a very slow component (release from GSats) [30]. Since 2-FA is an analog of epibatidine, it is speculated that this slow component of dissociation is similar between ligands and can be measured in vivo. Future work should explore whether an estimate of 2-FA release from GSats can be derived; however, the radioactive decay of F-18 may present a challenge when imaging for long scan durations. Alternatively, future studies could incorporate single-photon emission computed tomography (SPECT) scans using I-123 or I-125 labeled analogs of epibatidine to measure the release of ligand from GSats.

To directly compare to DVR estimates from tissue reference methods, *BP*_ND_ + 1 values from the 2TCM fit were compared to DVR from Logan graphical analysis [33] using a cerebellum reference tissue. For 2-FA, *BP*_ND_ + 1 was significantly higher than DVR for the thalamus and midbrain across all mouse groups. This difference is primarily due to the large *BP*_ND_ values present in the cerebellum, resulting in the Logan DVR underestimating the true binding potential. For Nifene, *BP*_ND_ + 1 and DVR estimates were within the same order of magnitude in the thalamus and midbrain for WT mice. This is likely because the levels of specific binding of Nifene in the cerebellum are much lower than observed with 2-FA (Figure 1b). Despite the lack of statistical significance, *BP*_ND_ + 1 values for the WT mice were higher than the observed DVR values, indicative of some levels of specific binding present in the cerebellum, resulting in DVR potentially underestimating the true binding potential. *BP*_ND_ + 1 values for KO and AN mice were significantly higher than DVR values for both the thalamus and midbrain; however, these mice display very low binding of ligand.

To test the reliability of the Python solver for the 2TCM fitting, simulations were performed on the 2-FA and Nifene PET data. Using a mix of male and female mice from the WT, KO, and AN groups, artificial Gaussian noise was applied to the TACs of the thalamus, cerebellum and left ventricle. The noise-induced TACs were then fit by the 2TCM and estimates of the rate constants and binding potential values were derived and compared to the true values derived from the noise-free data. For all groups of mice imaged with 2-FA, the true estimates of *BP*_ND_, *K*_1_, *k*_2_, *k*_3_, and *k*_4_ fell within the 25th and 75th percentiles of the simulated values. These findings suggest that the Python solver provides stable estimates of the radioligand rate constants for 2-FA. For Nifene, the true estimates of *BP*_ND_ fell within the 25th and 75th percentiles of the simulated values for all groups of mice; however, estimates of *K*_1_ and *k*_2_ were underestimated and *k*_3_ and *k*_4_ were overestimated for the WT group only. Since the Nifene TACs were well approximated by the 1TCM, the 2TCM fit had higher levels of uncertainty affiliated with the outcome measures. This uncertainty coupled with the slow input function kinetics resulting from the IP injection likely resulted in the poor estimation of the rate constants for Nifene. Future work will evaluate the 2TCM fit with 2-FA and Nifene delivered via IV injection to improve the ligand kinetics.

### Limitations to the Study

Limitations to the current study include not performing arterial sampling on the animals to confirm the left ventricle blood curve, not measuring for potential blood metabolites, which may influence the left ventricle radioactivity curve, and not exploring other brain tissue reference regions (such as the corpus collosum) to compare against the left ventricle blood curve due to the small size of the mouse brain and limited resolution of the PET scanner. While these corrections were not performed, our rate constant estimates were comparable to studies performed with the same radioligands in nonhuman primates [26,27], providing confidence that these results are suitable for exploratory analyses in mouse models of nicotine addiction. To further validate the use of a left ventricle image-derived input function, future studies incorporating arterial sampling through cannulation should be performed for comparison.

## 4. Materials and Methods

### 4.1. Animals

For 2-FA PET, 5 male and 5 female wild type (WT) mice, 8 male and 3 female β2 nAChR knockout (KO) mice, and 3 male and 3 female acute nicotine-treated (AN) mice were imaged. For Nifene PET, 7 male and 5 female WT mice, 5 male and 2 female KO mice, and 2 male and 2 female AN mice were imaged. The male and female KO mice and their WT littermates were generated in house by breeding a heterozygous pair on the C57BL/6J background purchased from the Jackson Lab (Bar Harbor, ME, USA) [34]. Additional male and female WT mice at the same age were purchased directly from the Jackson lab and used in the same manner as the WT littermates. Animals were housed in The University of Chicago Animal Research Resources Center. The Institutional Animal Care and Use Committee of the University of Chicago, in accordance with National Institutes of Health guidelines, approved all animal procedures. Mice were maintained at 22–24 °C on a 12:12-h light–dark cycle and provided food (standard mouse chow) and water ad libitum. All mice were 3–10 months old.

### 4.2. Radioligand Syntheses

Syntheses of both [^18^F]2-FA and [^18^F]Nifene were carried out at the Cyclotron Facility of The University of Chicago (Chicago, IL, USA). 2-FA was synthesized from the commercially available precursor, 2-TMA-A85380 (American Biochemicals Inc., College Station, TX, USA). Nifene was synthesized from the precursor N-BOC-nitroNifene. An IBA Synthera V2 automatic synthesis module (IBA, Louvain-la-Neuve, Belgium) equipped with Synthera preparative HPLC was used for the radiolabeling and purification inside a Comecer hot cell. The radiochemical yield of 2-FA was 34% (decay corrected, based on HPLC analysis of the crude product) with molar activities >111 GBq/μmol and radiochemical purity >99%. The radiochemical yield of Nifene was 6.3% (decay corrected) with molar activities >111 GBq/μmol and radiochemical purity >99%.

### 4.3. PET/CT Imaging

The imaging protocols were designed based on previous reports for 2-FA and Nifene [35]. An intraperitoneal (IP) catheter was placed at the lower right abdominal area of each mouse before imaging. The animal was then placed into the β-Cube preclinical microPET imaging system (Molecubes, Gent, Belgium) in a small animal holder. The ligand was delivered in 200 μL isotonic saline via the IP catheter and followed with addition of 100 μL of fresh saline. For AN mice, 0.5 mg/kg body weight of nicotine was injected IP 15 min before radioligand injection. Whole-body imaging was acquired with a 133 mm × 72 mm field of view (FOV) and an average spatial resolution of 1.1 mm at the center of the FOV [36]. List-mode data were recorded for 180 min for 2-FA and 60 min for Nifene followed by a reference CT image on the X-Cube preclinical microCT imaging system (Molecubes, Gent, Belgium). The images were reconstructed using an OSEM reconstruction algorithm that corrected for attenuation, randoms, and scatter with an isotropic voxel size of 400 μm. The re-binned frame rate for 2-FA was 10 × 60 s–17 × 600 s and the frame rate for Nifene was 12 × 10 s–18 × 60 s–8 × 300s. CT images were reconstructed with a 200 μm isotropic voxel size and used for anatomic co-registration, scatter correction, and attenuation correction. Animals were maintained under 1–2% isoflurane anesthesia in oxygen during imaging. Respiration and temperature were constantly monitored and maintained using the Molecubes monitoring interface and a Small Animal Instruments (SAII Inc., Stoney Brook, NY, USA) set up. All animals survived the imaging session.

### 4.4. Image Quantification

For each mouse imaged with 2-FA or Nifene, all PET frames were averaged and coregistered with the anatomical CT using VivoQuant (Invicro, Boston, MA, USA). The resulting transformations were then applied to each individual PET image frame to align the PET with the CT. Brain regional analysis was performed using a 3-dimensional mouse brain atlas available through VivoQuant, which is based on the Paxinos–Franklin atlas registered to a series of high-resolution magnetic resonance images with 100 µm near isotropic data that has been applied in other studies [37,38,39]. The brain atlas was warped into the CT image space and used for volume of interest (VOI) extraction of the whole brain, cerebellum, thalamus, and midbrain from the PET images. Regional radioactivity concentrations were converted into standardized uptake values (SUVs) by normalizing the signal by the injected dose of the radioligand and the body weight of the animal. In addition, a spherical VOI was hand-drawn over the left ventricle of the heart to act as an image-derived input function for the kinetic modeling analyses.

### 4.5. Radioligand Kinetic Modeling

Details on the parameters and methodology of implementing the two-tissue compartmental model (2TCM) are described fully in Appendix A. Using radioactivity time–activity curves (TACs) for all mice with the left ventricle TAC as an input function, the 2TCM was fit to the data to solve for the rate constants using an in-house Python software (Spyder v5.1.5). To test the validity of the Python software, the 2TCM was also applied to the data using the built-in solver provided through the π.PMOD software (v3.8). Taking the ratio of *k_3_* and *k_4_* resulted in an estimate of the binding potential (*BP*_ND_), which was determined for the thalamus, midbrain, and cerebellum. Previous imaging studies of 2-FA and Nifene utilized the distribution volume ratio (DVR) as the outcome measure of specific binding, which is related to *BP*_ND_ via the following formula:(1)$$DVR=BP_{ND}+1,$$

For the current study, the DVR was determined for the thalamus and midbrain using Logan graphical analysis [33] with the cerebellum acting as a tissue reference region. Using the concentration TACs generated from the VOIs from the VivoQuant brain atlas, a simplified reference tissue model (SRTM) [40] was used to compute the ${k_{2}}^{\prime }$ value for each region. Then, a Logan plot was created, and the slope used as the DVR.

### 4.6. 2TCM Simulations

To validate the repeatability of the in-house Python solver for the 2TCM, the 2TCM was solved for a series of simulated 2-FA and Nifene data. Details on the simulation methodology are described fully in Appendix A. For both 2-FA and Nifene, independently, regional TACs were taken from a single WT, KO, and AN mouse, and the 2TCM was used to determine the true values for the parameters *K*_1_, *k*_2_, *k*_3_, and *k*_4_. For each time point of the TAC, artificial noise was applied to the radioactivity concentration value with a noise level of 0.2 (noise standard deviation relative to the amplitude) such that the noise level was proportional to the amplitude/time duration. The 2TCM fit was then applied to the noise-induced thalamus and cerebellum TACs using the noise-induced left ventricle TAC as the input function. This process was repeated over 50 iterations for each mouse and radioligand. The parameters for the initial guess for each variable ranged as follows: *K*_1_, *k*_2_, *k*_3_, *k*_4_ = 0.0001–5, and *v_p_* = 0.0001–0.3. Values for *K*_1_, *k*_2_, *k*_3_, *k*_4_, and *BP*_ND_ were then compared against the true values derived for each mouse.

### 4.7. Statistical Analyses

All statistical analyses were performed using R v3.0 (The R Project for Statistical Computing). Normality of the data was assessed using the Shapiro–Wilk test. For WT, AN, and KO mice imaged with 2-FA or Nifene, the PET values did not significantly deviate from the normal distribution (all *p* > 0.05). Peak SUVs (max SUV across entire scan duration) and the mean SUVs from the final 30 min image frames of the left ventricle were compared between WT, KO, and AN mice using analysis of variance (ANOVA). For 2-FA and Nifene independently, *K*_1_ values from the thalamus and midbrain were compared between WT, KO, and AN mice using ANOVA with post hoc Tukey’s HSD test for multiple comparisons. The ANOVA and post hoc tests were then repeated for *k*_2_, *k*_3_, *k*_4_, and *BP*_ND_ values. For all ANOVA analyses, effect sizes are presented as η^2^ with 95% CIs. For both radioligands, the 2TCM rate constants from the thalamus and midbrain determined from the in-house Python solver were directly compared to the rate constants determined from PMOD for the WT, KO, and AN mice, independently using paired samples *t*-tests. *BP*_ND_ + 1 values determined from the 2TCM fit (left ventricle input function) were then directly compared with DVR values from Logan analysis (cerebellum reference tissue) using paired samples *t*-tests.

## 5. Conclusions

In summary, these exploratory results show that use of an image-derived input function is suitable for quantification of 2-FA and Nifene PET data in the study of mouse models of nicotine addiction. Consistent with the previous rodent work performed with these radioligands, both 2-FA and Nifene display low levels of specific binding in the cerebellum, further emphasizing the need for analyses using an arterial input function to better estimate α4β2R density. Due to the similarity in structure and kinetics of 2-FA and Nifene to varenicline and nicotine, these radioligands are ideal for studying the underlying mechanisms of nicotine addiction and smoking cessation in vivo.

## Figures and Tables

**Figure 1 ijms-24-15510-f001:**
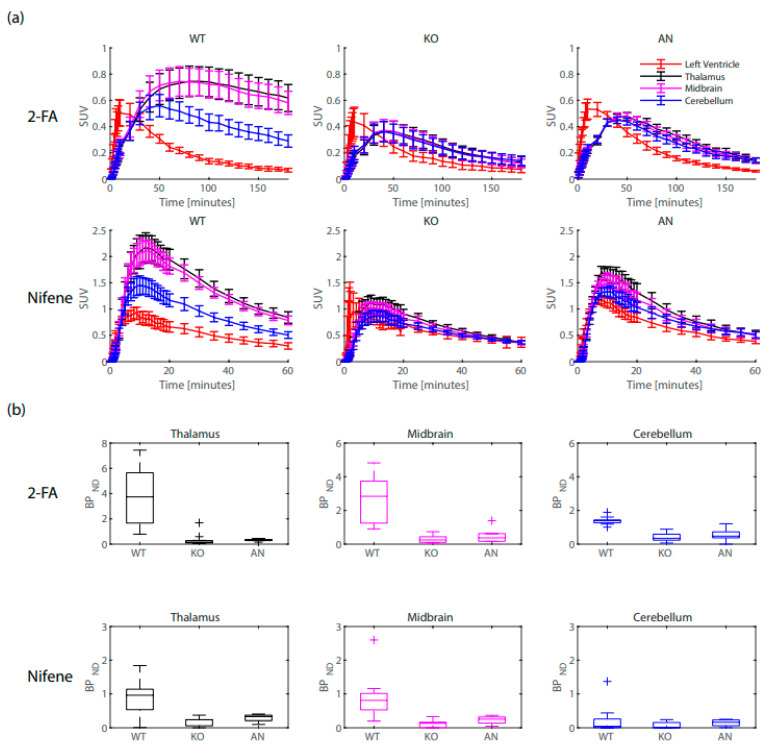
(**a**) 2-FA and Nifene standardized uptake value (SUV) time-activity curves (TACs) for WT, KO, and AN mice. Error bars represent the standard deviation. (**b**) 2-FA and Nifene *BP*_ND_ values for the thalamus (black), midbrain (magenta) and cerebellum (blue) of WT, KO, and AN mice.

**Figure 2 ijms-24-15510-f002:**
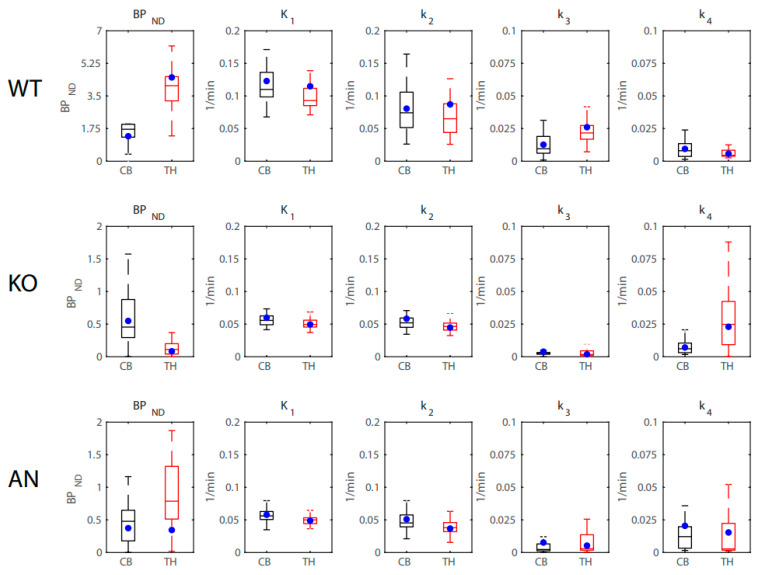
Distribution of simulated 2-FA *BP*_ND_, *K*_1_, *k*_2_, *k*_3_, and *k*_4_ estimates from WT, KO, and AN mice in the cerebellum (CB; black) and thalamus (TH; red) using a left ventricle input function. The true values of each estimate are displayed as blue circles.

**Figure 3 ijms-24-15510-f003:**
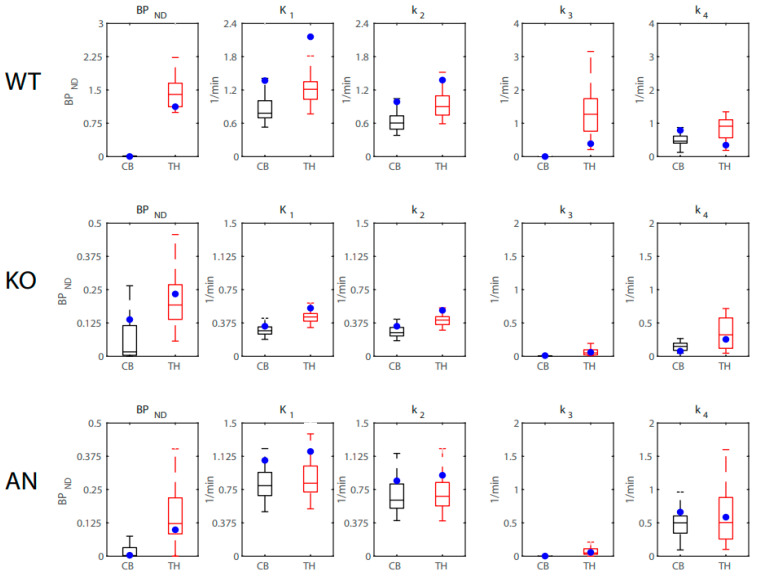
Distribution of simulated Nifene *BP*_ND_, *K*_1_, *k*_2_, *k*_3_, and *k*_4_ estimates from WT, KO, and AN mice in the cerebellum (CB; black) and thalamus (TH; red) using a left ventricle input function. The true values of each estimate are displayed as blue circles.

**Table 1 ijms-24-15510-t001:** Rate constants for WT, KO, and AN mice imaged with 2-FA and Nifene with ANOVA statistics and effect size (η^2^) estimates with 95% Cis.

Radioligand	Region	Rate Constant (1/min)	WT	KO	AN	ANOVA F(df)	*p*-Value	η^2^
2-FA	Thalamus	*K_1_*	0.098 (0.013)	0.066 (0.0061)	0.060 (0.0036)	3.88(2)	0.035	0.24 [0.00, 0.45]
		*k_2_*	0.075 (0.015)	0.055 (0.0066)	0.050 (0.0046)	1.37(2)	0.27	0.10 [0.00, 0.31]
		*k_3_*	0.029 (0.0064)	0.0049 (0.0016)	0.0054 (0.00080)	9.75(2)	0.00080	0.45 [0.11, 0.62]
		*k_4_*	0.0077 (0.0013)	0.020 (0.0037)	0.016 (0.0017)	5.48(2)	0.011	0.31 [0.02, 0.51]
	Midbrain	*K_1_*	0.11 (0.017)	0.076 (0.0087)	0.062 (0.0051)	2.67(2)	0.090	0.18 [0.00, 0.39]
		*k_2_*	0.076 (0.020)	0.066 (0.0087)	0.051 (0.0055)	0.54(2)	0.59	0.04 [0.00, 0.21]
		*k_3_*	0.024 (0.0062)	0.0060 (0.0012)	0.0031 (0.00070)	6.70(2)	0.0049	0.36 [0.05, 0.55]
		*k_4_*	0.0084 (0.0016)	0.026 (0.0037)	0.0093 (0.0022)	11.70(2)	0.00029	0.49 [0.16, 0.65]
	Cerebellum	*K_1_*	0.10 (0.016)	0.077 (0.0072)	0.063 (0.0053)	2.53(2)	0.10	0.17 [0.00,0.39]
		*k_2_*	0.080 (0.016)	0.068 (0.0069)	0.058 (0.0057)	0.68(2)	0.52	0.05 [0.00, 0.23]
		*k_3_*	0.011 (0.0025)	0.0056 (0.00083)	0.0036 (0.0010)	4.29(2)	0.026	0.26 [0.00, 0.47]
		*k_4_*	0.0081 (0.0017)	0.017 (0.0040)	0.015 (0.0059)	1.534(2)	0.24	0.11 [0.00, 0.31]
Nifene	Thalamus	*K_1_*	1.66 (0.24)	0.55 (0.089)	1.25 (0.072)	7.82(2)	0.0027	0.42 [0.08, 0.60]
		*k_2_*	1.23 (0.26)	0.51 (0.074)	1.09 (0.068)	3.05(2)	0.068	0.22 [0.00, 0.44]
		*k_3_*	0.43 (0.11)	0.047 (0.011)	0.14 (0.024)	4.96(2)	0.017	0.31 [0.01, 0.52]
		*k_4_*	0.50 (0.12)	0.27 (0.034)	0.51 (0.12)	1.71(2)	0.20	0.13 [0.00, 0.35]
	Midbrain	*K_1_*	1.94 (0.31)	0.50 (0.080)	1.28 (0.099)	8.00(2)	0.0024	0.42 [0.08, 0.60]
		*k_2_*	1.52 (0.35)	0.47 (0.069)	1.14 (0.061)	3.54(2)	0.047	0.24 [0.00, 0.46]
		*k_3_*	0.50 (0.13)	0.029 (0.0085)	0.12 (0.026)	5.75(2)	0.0098	0.34 [0.03, 0.54]
		*k_4_*	0.60 (0.14)	0.26 (0.046)	0.60 (0.077)	2.21(2)	0.13	0.17 [0.00, 0.39]
	Cerebellum	*K_1_*	1.56 (0.33)	0.33 (0.061)	0.91 (0.10)	5.19(2)	0.014	0.32 [0.02, 0.52]
		*k_2_*	1.19 (0.32)	0.31 (0.053)	0.83 (0.072)	2.81(2)	0.082	0.20 [0.00, 0.42]
		*k_3_*	0.10 (0.055)	0.010 (0.0052)	0.033 (0.013)	1.21(2)	0.32	0.10 [0.00, 0.31]
		*k_4_*	0.62 (0.18)	0.19 (0.046)	0.34 (0.14)	2.00(2)	0.16	0.16 [0.00, 0.37]

**Table 2 ijms-24-15510-t002:** 2-FA rate constants measured using the in-house Python solver and the PMOD solver compared using pairwise *t*-tests.

Group	Rate Constant	Region	2TCM Python	2TCM PMOD	T-Value	*p*-Value
WT	*K_1_*	Thalamus	0.098 (0.013)	0.13 (0.022)	1.17	0.27
		Midbrain	0.11 (0.017)	0.12 (0.014)	1.23	0.25
	*k_2_*	Thalamus	0.075 (0.015)	0.19 (0.085)	1.24	0.24
		Midbrain	0.076 (0.020)	0.13 (0.028)	1.61	0.14
	*k_3_*	Thalamus	0.029 (0.0064)	0.071 (0.021)	1.69	0.12
		Midbrain	0.024 (0.0062)	0.043 (0.0076)	2.26	0.051
	*k_4_*	Thalamus	0.0077 (0.0013)	0.026 (0.017)	0.98	0.35
		Midbrain	0.0084 (0.0016)	0.013 (0.0032)	1.28	0.23
	*BP* _ND_	Thalamus	3.84 (0.70)	5.70 (1.25)	1.42	0.19
		Midbrain	2.66 (0.41)	3.78 (0.55)	1.94	0.084
KO	*K_1_*	Thalamus	0.066 (0.0061)	0.14 (0.030)	2.39	0.038
		Midbrain	0.076 (0.0087)	0.18 (0.041)	2.46	0.034
	*k_2_*	Thalamus	0.055 (0.0066)	0.44 (0.18)	1.97	0.077
		Midbrain	0.066 (0.0087)	0.54 (0.20)	2.18	0.054
	*k_3_*	Thalamus	0.0049 (0.0016)	0.090 (0.023)	3.49	0.0059
		Midbrain	0.0060 (0.0012)	0.092 (0.027)	2.98	0.014
	*k_4_*	Thalamus	0.020 (0.0037)	0.049 (0.0060)	5.39	0.00030
		Midbrain	0.026 (0.0037)	0.044 (0.0080)	3.02	0.013
	*BP* _ND_	Thalamus	0.34 (0.14)	2.07 (0.63)	2.44	0.035
		Midbrain	0.29 (0.064)	1.92 (0.54)	2.66	0.024
AN	*K_1_*	Thalamus	0.060 (0.0036)	0.34 (0.13)	1.85	0.12
		Midbrain	0.062 (0.0051)	0.37 (0.12)	2.26	0.073
	*k_2_*	Thalamus	0.050 (0.0046)	1.23 (0.63)	1.71	0.15
		Midbrain	0.051 (0.0055)	1.19 (0.47)	2.20	0.079
	*k_3_*	Thalamus	0.0054 (0.00080)	0.087 (0.023)	2.91	0.033
		Midbrain	0.0031 (0.00070)	0.085 (0.028)	2.67	0.045
	*k_4_*	Thalamus	0.016 (0.0017)	0.030 (0.0028)	3.14	0.026
		Midbrain	0.0093 (0.0022)	0.031 (0.0024)	6.03	0.0018
	*BP* _ND_	Thalamus	0.33 (0.037)	2.84 (0.86)	2.66	0.045
		Midbrain	0.52 (0.17)	2.61 (0.77)	2.77	0.039

**Table 3 ijms-24-15510-t003:** Nifene rate constants measured using the in-house Python solver and the PMOD solver compared using pairwise *t*-tests.

Group	Rate Constant	Region	2TCM Python	2TCM PMOD	T-Value	*p*-Value
WT	*K_1_*	Thalamus	1.66 (0.24)	1.18 (0.13)	3.60	0.0042
		Midbrain	1.94 (0.31)	1.45 (0.17)	2.83	0.016
	*k_2_*	Thalamus	1.23 (0.26)	0.61 (0.12)	3.37	0.0062
		Midbrain	1.52 (0.35)	0.85 (0.15)	2.33	0.040
	*k_3_*	Thalamus	0.43 (0.11)	0.76 (0.51)	0.72	0.49
		Midbrain	0.50 (0.13)	0.36 (0.12)	0.68	0.51
	*k_4_*	Thalamus	0.50 (0.12)	1.63 (0.62)	2.09	0.060
		Midbrain	0.60 (0.14)	1.42 (0.56)	1.77	0.10
	*BP* _ND_	Thalamus	0.90 (0.14)	0.57 (0.18)	2.28	0.043
		Midbrain	0.88 (0.17)	0.63 (0.22)	1.10	0.30
KO	*K_1_*	Thalamus	0.55 (0.089)	0.52 (0.12)	0.73	0.48
		Midbrain	0.50 (0.080)	0.50 (0.11)	0.0096	0.99
	*k_2_*	Thalamus	0.51 (0.074)	0.63 (0.29)	0.52	0.62
		Midbrain	0.47 (0.069)	0.45 (0.098)	0.33	0.75
	*k_3_*	Thalamus	0.047 (0.011)	0.075 (0.037)	0.62	0.55
		Midbrain	0.029 (0.0085)	0.068 (0.043)	0.87	0.41
	*k_4_*	Thalamus	0.27 (0.034)	2.64 (1.04)	2.16	0.062
		Midbrain	0.26 (0.046)	0.65 (0.35)	1.01	0.34
	*BP* _ND_	Thalamus	0.18 (0.038)	0.030 (0.010)	4.13	0.0033
		Midbrain	0.12 (0.034)	0.11 (0.047)	0.15	0.88
AN	*K_1_*	Thalamus	1.25 (0.072)	1.00 (0.050)	1.92	0.15
		Midbrain	1.28 (0.099)	1.25 (0.083)	0.20	0.86
	*k_2_*	Thalamus	1.09 (0.068)	0.86 (0.15)	1.09	0.35
		Midbrain	1.14 (0.061)	1.29 (0.21)	0.57	0.61
	*k_3_*	Thalamus	0.14 (0.024)	0.65 (0.39)	1.19	0.32
		Midbrain	0.12 (0.026)	1.37 (1.04)	1.02	0.38
	*k_4_*	Thalamus	0.51 (0.12)	4.23 (1.73)	1.90	0.15
		Midbrain	0.60 (0.077)	3.83 (1.62)	1.80	0.17
	*BP* _ND_	Thalamus	0.29 (0.056)	0.31 (0.17)	0.11	0.91
		Midbrain	0.23 (0.056)	0.40 (0.14)	0.88	0.44

**Table 4 ijms-24-15510-t004:** Pairwise comparisons between *BP*_ND_ + 1 from the 2TCM and DVR from Logan graphical analysis for 2-FA and Nifene.

Radioligand	Group	Region	*BP*_ND_ + 1	DVR	T-Value	*p*-Value
2-FA	WT	Thalamus	4.84 (0.70)	2.35 (0.15)	3.94	0.0034
		Midbrain	3.66 (0.41)	1.96 (0.080)	4.15	0.0025
	KO	Thalamus	1.34 (0.14)	0.97 (0.019)	2.57	0.028
		Midbrain	1.29 (0.064)	0.97 (0.011)	4.78	0.00075
	AN	Thalamus	1.33 (0.037)	1.15 (0.051)	2.73	0.042
		Midbrain	1.52 (0.17)	1.05 (0.023)	2.40	0.062
Nifene	WT	Thalamus	1.90 (0.14)	1.79 (0.079)	0.65	0.53
		Midbrain	1.88 (0.17)	1.54 (0.092)	1.82	0.095
	KO	Thalamus	1.18 (0.038)	0.94 (0.039)	5.03	0.0010
		Midbrain	1.12 (0.034)	0.98 (0.015)	3.11	0.015
	AN	Thalamus	1.29 (0.056)	0.93 (0.016)	5.79	0.010
		Midbrain	1.23 (0.056)	0.99 (0.0049)	3.57	0.037

## Data Availability

Data can be made available upon request.

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
