# Peer review of "Evaluation of an Image-Derived Input Function for Kinetic Modeling of Nicotinic Acetylcholine Receptor-Binding PET Ligands in Mice"

_ijms, 2023, doi:10.3390/ijms242115510_

Round 1

Reviewer 1 Report

Thank you for the opportunity to review the manuscript "Evaluation of kinetic modeling strategies for nicotinic acetylcholine receptor-binding PET ligands".  This manuscript tries to validate of image-derived input function for 2TCM compared to the reference tissue method in mice PET study using two tracers for alpha2-beta4 nicotinic actylcholine receptors. The authors sincerely and specifically described, and the trial is interesting. However, the reviewer can not accept the reliability of most of the results, then the reviewer can not agree with the claims of the manuscripts.

Major comments:

(1) The reviewer can not accept the reliability of the results because of the reasons described below. 

(1-1) The curves of whole blood can not be used as the input function for 2TCM.

Theoretically, the estimation of BP_ND using 2TCM could be considered as the separations of the components of C_ND(t) and C_S(t) from measured C_T(t). One of the main features used for the estimation is the shape of the measured (C_T(t)) and simulated (C_ND(t) and C_S(t)) curves. The simulated curves are affected by the input function and they should be generated using metabolite-corrected plasma input curves. However, the authors only have the image-derived whole blood curves from the left ventricle, and the shape of the curve would be different from the curves of metabolite-corrected plasma because the rate of the un-metabolites would depend on the time after injection.  As the authors might be experienced in the comparisons between Python and PMOD, the estimations of 2TCM are very sensitive and unstable, then these differences in the input functions would cause huge alternations to the estimated results. The use of the whole blood input function, as an alternative to the metabolite-corrected plasma input function can not be accepted before the validations compared with the results of 2TCM using the measured metabolite-corrected plasma input function on each PET tracer and each target species. The review article (ref. 27) referred in the manuscripts which concludes "it is also a challenging technique that can be successfully implemented in clinical practice only for a small number of tracers." did not support this study.

(1-2) The estimated rate constants over 1 were included.

The rate constants (K_1, k_2, k_3, and k_4) in Table 1, 2, and 3, include the estimated results over 1. The rate constants must be within 0 to 1, and the results of over 1 indicate the estimations of 2TCM failed, and the conditions (such as the initial values of the rate constants) of the whole estimations (including for the curves which all estimated k-parameters were within 0 to 1) should be changed. Therefore, the results on the Tables and calculated BP_NDs can not be accepted as reliable.

(2) The results would indicate the reference tissue method is better than the 2TCM in this study.

From Table 4, the SDs of DVR from Logan graphical analysis were much smaller than those of BPND from 2TCM. This is a very attractive advantage because the number of samples could decrease in the actual applications such as for the validation of the significant differences.

As the authors mentioned, specific binding in the cerebellum and the underestimation of BP_NDs using the reference tissue method could not be negligible. However, in the reviewer's opinion, the underestimations would be limited (especially for [18F]Nifene) or could be included in the interpretations of the results for most of the applications. For these situations (rodent PET studies using these two tracers), the adverse effects from other candidates, such as the more variable results of BP_ND from the 2TCM estimations, the effect of anesthesia, the peripheral changes by the injection of the target drugs (such as nicotine for AN mice in this manuscript), and the species differences between rodents and human, would be much larger than those of specific binding in the reference tissue or underestimations of BP_ND.

(3) The authors should stop over-interpreting for results with non-significant differences.

The t-test can be interpreted only when the test reveals the result with significant differences. The result with non-significant differences did NOT indicate the null hypothesis (the difference between the results was small, acceptable, or negligible) is true. The "no significant differences" used in the discussion would mainly be caused by the large variabilities in the estimation or estimated results of 2TCM.

Minor comments:

(1) Title is not adequate. Sentence like "image-derived input functions"  and "PET study in mouse" should be included.

(2) The representative PET images (early and latter phase) and reference CT images (with ROIs) of the slices including the brain and the left ventricle, should be figured.

(3) The results of the cerebellum should be included in Table 1, 2, and 3.

(4) The comparison of BP_ND should be included in Table 2 and 3.

(5) The initial values of the parameters (K_1, k_2, k_3, k_4, and v_p) in the estionation of 2TCM should be described in the main text.

(6) The "goodness-of-fit" in 2TCM such as time-courses of residuals or Akaike Information Criterion, should be discussed.

(7) The first part of section 4.5 is slightly text-bookish. The equations 1 and 2, and the explanations should be moved to the Appendix.

(8) The valuables or parameters (such as K_1, k_2,...) should be noted using italic type (except DVR) not only in the equations but also in the main text and Tables. On the other hand, the characters in the subscripts of the valuables or parameters (such as P in C_P and ND in BP_ND) should be noted using Roman type. Please refer to the consensus nomenclature of this research area [1].

(9) l.336 (p.11) wild type mice (WT) -> wild type (WT) mice

(10) The name of the journal of ref 22 is missing, and the reference could be the pre-print(biorxiv) of ref 14.

[1] Innis et al, Consensus Nomenclature for in vivo Imaging of Reversibly Binding Radioligands, J Cereb Blood Flow Metab, 27: 1533-9, 2007. doi: 10.1038/sj.jcbfm.9600493

Reviewer 2 Report

The study investigates the kinetics of two radiotracers, 2-FA and Nifene, using an image-derived arterial input function in PET imaging. While the experiment is technically sound, there are concerns about the novelty of the research, which limits its overall impact.

Major Comment: The major concern with this study is its limited novelty. The PET imaging of 2-FA and Nifene has already been covered in a previous paper by the same authors (Hannah J. Zhang et al., J Neurosci. 2023). The primary improvement in this study lies in the use of the left ventricle as an image-derived input function instead of the cerebellum as a reference. While this change is reasonable, the use of an arterial input function is not new in PET imaging studies. Moreover, it's important to note that in brain PET studies, the left ventricle is often not in the field of view. Therefore, the authors should emphasize what truly distinguishes this study from their previous work and from existing literature to clarify its contribution to the field.

Minor Comment: In the "Simulations" result section, the true values and simulated estimates are confusing. It needs a better explanation how they are generated. 

Reviewer 3 Report

In the article "Evaluation of kinetic modeling strategies for nicotinic acetyl-2-choline receptor-binding PET ligands", the authors evaluated nicotine-induced upregulation of α4β2 receptors by quantifying PET images (time-activity curve acquisition) of two specific ligands (2-FA and nifene) in the left ventricle of mice, using the cerebellum as a tissue reference region. The article is well written with an adequate methodology using appropriate controls: wild-type, β2-knockout, and acute nicotine-treated mice to explore how different mouse models influence the radiotracer activity concentration in the blood pool. The results are compelling, showing significant differences for 2-FA and nifene between wild-type, β2-knockout, and acute nicotine-treated mice for K1 values in the thalamus but not in the midbrain, and no significant differences between SUV TACs of the left ventricle (an appropriate reference for the mouse model), as well as the use of the Python solver and PMOD for two-tissue compartmental model fitting of the PET data. The authors have demonstrated that 2-FA and nifene radiotracers are potentially useful for studying the underlying mechanisms of nicotine addiction and smoking cessation in vivo. Therefore, I recommend publication of the manuscript as is.

Author Response

We thank the reviewers for their feedback to improve the manuscript. We have thoughtfully addressed each point below in bold.

Reviewer 3:

In the article "Evaluation of kinetic modeling strategies for nicotinic acetyl-2-choline receptor-binding PET ligands", the authors evaluated nicotine-induced upregulation of α4β2 receptors by quantifying PET images (time-activity curve acquisition) of two specific ligands (2-FA and nifene) in the left ventricle of mice, using the cerebellum as a tissue reference region. The article is well written with an adequate methodology using appropriate controls: wild-type, β2-knockout, and acute nicotine-treated mice to explore how different mouse models influence the radiotracer activity concentration in the blood pool. The results are compelling, showing significant differences for 2-FA and nifene between wild-type, β2-knockout, and acute nicotine-treated mice for K1 values in the thalamus but not in the midbrain, and no significant differences between SUV TACs of the left ventricle (an appropriate reference for the mouse model), as well as the use of the Python solver and PMOD for two-tissue compartmental model fitting of the PET data. The authors have demonstrated that 2-FA and nifene radiotracers are potentially useful for studying the underlying mechanisms of nicotine addiction and smoking cessation in vivo. Therefore, I recommend publication of the manuscript as is.

We thank the reviewer for their approval of the manuscript in its current state.

Round 2

Reviewer 1 Report

Please refer to the attached Word file.

Author Response

Response to Referee 1:

We again thank the reviewer for their thorough evaluation of this manuscript. However, we respectfully disagree with the claims brought forth regarding the validity of our methodology and results.

  1. Regarding use of a whole blood curve as an input function, we have added more references with various radioligands in the introduction showing that these types of studies are feasible:

“Studies incorporating image-derived input functions have shown that the time-activity curve (TAC) from the left ventricle of the heart was an accurate representation of arterial blood, obviating the need for arterial cannulation[21–25].

Here, we provide an exploratory analysis of the use of the left ventricle as an image-derived input function without arterial sampling for quantification of 2-FA and Nifene PET images of mice…”

And in the discussion:

“An ROI of the left ventricle was chosen as the image-derived input function due to its accurate representation of the plasma fraction as shown in other studies[20–25], and in its current application, the left ventricle TAC showed rapid uptake and rapid clearance of 2-FA and Nifene in the blood pool. Across WT, KO and AN mice, no differences in shape or radioactivity concentration were observed between the SUV TACs of the left ventricle, suggesting the left ventricle blood curve is stable across different mouse models studied in nicotine addiction and can be a suitable reference for these mouse models when imaging with 2-FA and Nifene.”

And, in our limitations to the study section:

“While these corrections were not performed, our rate constant estimates were comparable to studies performed with the same radioligands in nonhuman primates[26,27], providing confidence that these results are suitable for exploratory analyses in mouse models of nicotine addiction.”

Furthermore, we emphasize that this is an exploratory analysis to evaluate whether this type of methodology can be achieved specifically for applications in mouse models of nicotine addiction, and do not speculate its feasibility for use beyond rodent imaging.

  1. The reviewer claims that because rate constant estimates exceed 1.0 (1/min), that the results of the 2TCM fit are invalid.

The assumption that K1 values exceeding 1.0 (1/min) is a result of a poor model fit is incorrect. Values over 1.0 (1/min) indicate that complete extraction of the radioligand from plasma to tissue is observed. This is common when active transporters facilitate transport across the blood-brain barrier, resulting in very rapid kinetics and high rate constant values. In our study, only Nifene has rate constant values that exceed 1.0 (1/min), and there is a wealth of literature that suggests the rapid BBB permeability is a result of α4β2R ligand interaction with amine transporters. Furthermore, Nifene has been evaluated in nonhuman primate models with the 2TCM fit that include arterial sapling and metabolite correction. It has been shown that rate constants with these corrections exceed values of 1.0 (1/min), matching our values in mice without corrections in place. We have emphasized these points in the discussion as follows:

“While this study is the first to assess kinetic modeling strategies of α4β2R ligands in mouse models of nicotine addiction, other studies have evaluated the kinetics in nonhuman primates[26,27]. For rhesus macaques, modeling of dynamic Nifene PET data was performed with arterial sampling and metabolite correction[26,27]. BPND values in cortical and subcortical regions of known Nifene binding were comparable between the rhesus macaques[26] and the mice used in our current study. Another study in rhesus macaques performed a 2TCM fit of the PET data with metabolite correction and found that average K1 values for Nifene were above 1.0 (1/min), confirming the high K1 values observed in our mouse models without arterial sampling and metabolite correction. K1 values that exceed 1.0 (1/min) indicate complete extraction of the radioligand from plasma to tissue, consistent with what is observed with Nifene[27]. It is speculated that a transport mechanism is at play, specifically Nifene interaction with the blood-brain barrier amine transporter[28], which may account for the fast uptake rates of Nifene compared to 2-FA[27]. Similar α4β2R ligands with rapid in vivo kinetics, such as Flubatine (formerly NCFHEB), have been shown to interact with the blood-brain barrier amine transporter[29]. Furthermore, work in the field is ongoing to investigate the role of the amine transporter on Nifene passage across the blood-brain barrier[27].”

  1. “The results with non-significant differences” do not indicate “the results are similar”. From the results, the reviewer finds quite large differences, and can not accept the discussions that the results are similar.

We have adjusted the language in the discussion regarding the results as follows:

“Importantly, estimates calculated from the Python solver and PMOD were of within the same order of magnitude when fitting data from the WT mice, which displayed high levels of radioligand binding.”

“For Nifene, BPND+1 and DVR estimates were within the same order of magnitude in the thalamus and midbrain for WT mice.”